# Evaluation of Land Use and Land Cover Change and Its Drivers in Battambang Province, Cambodia from 1998 to 2018

Taingaun Sourn [1,*], Sophak Pok [2], Phanith Chou [3], Nareth Nut [1,4], Dyna Theng [4], Phanna Rath [2], Manuel R. Reyes [5] and P.V. Vara Prasad [5]

1     Graduate School, Royal University of Agriculture, Phnom Penh 12400, Cambodia; nnareth@rua.edu.kh
2     Faculty of Land Management and Land Administration, Royal University of Agriculture,
      Phnom Penh 12400, Cambodia; poksophak@gmail.com (S.P.); rathphanna.lmla@gmail.com (P.R.)
3     Faculty of Development Studies, Royal University of Phnom Penh, Phnom Penh 12150, Cambodia;
      chou.phanith@rupp.edu.kh
4     Faculty of Agricultural Engineering, Royal University of Agriculture, Phnom Penh 12400, Cambodia;
      thdyna@rua.edu.kh
5     Sustainable Intensification Innovation Lab (SIIL), Kansas State University, Manhattan, KS 66506, USA;
      mannyreyes@ksu.edu (M.R.R.); vara@ksu.edu (P.V.V.P.)
*     Correspondence: sourntaingauns@gmail.com; Tel.:+855-15-455-686

**Abstract:** The main objective of this research was to evaluate land use and land cover (LULC) change in Battambang province of Cambodia over the last two decades. The LULC maps for 1998, 2003, 2008, 2013 and 2018 were produced from Landsat satellite imagery using the supervised classification technique with the maximum likelihood algorithm. Each map consisted of seven LULC classes: built-up area, water feature, grassland, shrubland, agricultural land, barren land and forest cover. The overall accuracies of the LULC maps were 93%, 82%, 94%, 93% and 83% for 1998, 2003, 2008, 2013 and 2018, respectively. The LULC change results showed a significant increase in agricultural land, and a large decrease in forest cover. Most of the changes in both LULC types occurred during 2003–2008. Overall, agricultural land, shrubland, water features, built-up areas and barren land increased by 287,600 hectares, 58,600 hectares, 8300 hectares, 4600 hectares and 1300 hectares, respectively, while forest cover and grassland decreased by 284,500 hectares and 76,000 hectares respectively. The rate of LULC changes in the upland areas were higher than those in the lowland areas of the province. The main drivers of LULC change identified over the period of study were policy, legal framework and projects to improve economy, population growth, infrastructure development, economic growth, rising land prices, and climate and environmental change. Landmine clearance projects and land concessions resulted in a transition from forest cover and shrubland to agricultural land. Population and economic growth not only resulted in an increase of built-up area, but also led to increasing demand for agricultural land and rising land prices, which triggered the changes of other LULC types. This research provides a long-term and detailed analysis of LULC change together with its drivers, which is useful for decision-makers to make and implement better policies for sustainable land management.

**Keywords:** land use and land cover change; drivers; Cambodia; LULC; multi-temporal

## 1. Introduction

Land use and land cover (LULC) change happens due to natural and anthropogenic activities. One third of the Earth's land surface is currently being used for growing crops or grazing cattle [1]. The rapid growth of human population, infrastructure development, urban growth, and legal framework or policy change, in particular land tenure security and expansion of agricultural land, have occurred at the expense of natural land cover such as forests, shrubland, and grassland that provide valuable habitats for diverse terrestrial plant and animal species and ecosystem services for humankind [2–7]. Although LULC changes

may promote social and economic development, these changes also cause problems such global, regional and local climate change, land degradation, increased natural disasters, loss of wildlife habitat and biodiversity, water flow change, food insecurity and poor human health [8–12]. As global demand for food, fiber and biofuels reaches unprecedented levels, the supply of available land continues to decline [13]. Most of this land is concentrated in tropical forest regions, fueling the debate on how to reconcile the need for agricultural production with the conservation of forests and the maintenance of ecosystem services such as carbon storage, climate regulation and conservation biodiversity conservation [14].

LULC includes studies of deforestation, the expansion and intensification of agriculture, the energy footprint and urban growth [12]. Winkler et. al., [15] reported that in just six decades from 1960 to 2019, land use change has affected nearly a third of world land areas (32%). Therefore, it is about four times larger than previously estimated from the assessments of long-term land change. According to Macedo et. al. [14], deforestation in the Amazon border state of Mato Grosso decreased to thirty percent (30%) of its historical average from 1996 to 2005, while agricultural production reached a high record.

In Cambodia, LULC has dramatically changed over the last three decades following major political change, population and economic growth, market demand for food crops, climate change and human migration [16–18]. For instance, agricultural land expanded from 26% to 31% between 1997 and 2007 [19]. With increased demand for farmland, forest cover decreased through conversion to agricultural land for cropping [20]. Battambang province, located in northwest of Cambodia, is one of the key regions of economical, geographical and political importance. Battambang has a border with Thailand, which helps its import and export trade with neighboring countries. Most of the agricultural products, such as the rice (*Oryza sativa*), cassava (*Manihot esculenta*) and corn (*Zea mays*) grown in the province, are exported to Thailand. According to the provincial report [21], the provincial GDP was 1,460 million US dollars in 2016 (7.3% of the country's GDP). Geographically, Battambang province consists of a mixture of diverse LULC types including forests, agricultural land, built-up areas, grassland, shrubland, water bodies, and barren land [22]. The province has the largest cultivated land area, accounting for roughly 59% of the total provincial area [23]. The LULC has changed vastly and rapidly since the civil war ended as a result of the adoption of the win-win policy in 1998. Under that policy, Khmer Rouge forces and people including soldiers were guaranteed the right to life, jobs and profession for dignified living and protection of all their assets [24]. They were permitted to resettle, given responsibilities for land management and were allocated farming plots [25].

According to the Ministry of Agriculture, Forestry and Fisheries (MAFF) [26], deforestation in Cambodia caused a notable loss of forest cover from 10.83 million ha (59.64%) in 2006 to 8.52 million ha (46.90%) in 2014, and to 8.22 million ha (45.26%) in 2016. During this period, the area of arable land (e.g., fields, crops, gardening), plantations of rubber (*Hevea brasiliensis*) and oil palm (*Elaeis guineensis*) increased by about 2.69 million ha. Moreover, the Ministry of Environment (MoE), reported that, based on the forest assessment in 2016 [27], Cambodia's forest cover has decreased from 73.04% in 1965 to 48.14% in 2016. This was mainly caused by civil war, population growth, demand for agricultural land and other related factors. In 2016, the country's forest cover was about 8,742,401 hectares (48.14%), and the average annual loss rate of forest from 2014 to 2016 was 121,328 hectares (0.67%). Lohani et al. [28], as cited in [29], highlighted that from 1993 to 2017, the forest loss in Cambodia was 1,715,000 hectares, while the Tonle Sap Basin, in which total area of forest loss was small at 194,400 hectares, had the highest percentage of total forest loss of all study areas. In the western part of Cambodia, many large patches of forest in highlands (e.g., Cardamom Mountains) were lost. Unlike the northwestern part of Cambodia along the border with Thailand, a concentrated area of forest loss occurred in the early 2000s. It decreased only after nearly all remaining primary forests were lost. Kong et al. [20] reported that the total forest coverage (dense and degraded forests) remained almost unchanged between 1976 and 1997, which accounted for about 90% of the study area. However, about 13% of the dense forest area was degraded forestland. From 1997 to 2016, forest cover was reduced

dramatically to 25% in 2016, especially along the main roads. Around 65% of the forest cover loss occurred between 2006 and 2016 [20]. Moreover, Nut et al. [29] also reported that in Stung Sangkae catchment, which covers almost one-third of the Battambang province, the forest cover (such as evergreen, deciduous and mixed forest) occupied 43% in 2002 and decreased to 30% in 2015 due to the increase of agricultural land and paddy rice fields in the catchment.

Most of the research findings on LULC changes were carried out partially in Battambang province, as a complete picture of LULC changes in the whole province after the end of civil war in 1998 has not been well understood and documented. It is critical to understand these changes in depth to develop better land use policies and manage natural resources. Therefore, the goal of this research was to assess LULC change from 1998 to 2018. Specifically, the study attempts to ascertain and answer three main questions: (i) how has LULC changed from 1998 to 2018 (ii) what are the major drivers that caused the LULC change? and (iii) was the change different between the upland and lowland areas? The results of this study will provide important evidence and deep insights into LULC change over the past 20 years (1998–2018), providing a better long-term understanding of drivers and impacts of LULC changes. Furthermore, the findings can be used as information for decision-makers, policymakers and spatial planners to proactively direct future development and conservation to ensure the sustainability, inclusion and resilience of Battambang province.

## 2. Materials and Methods

### 2.1. Study Area

This study was conducted in Battambang province, the fifth-largest province in Cambodia (Figure 1). Located in the northwest part of the country, Battambang is characterized by four ecological zones: upland area, semi-upland area, lowland area and floodplain along Tonle Sap Lake, and has as a mixture of various land uses. The average elevation of the upland is about 118 meters above mean sea level (MSL) in the mountainous area and the average elevation of the lowland is approximately 9 MSL [30].

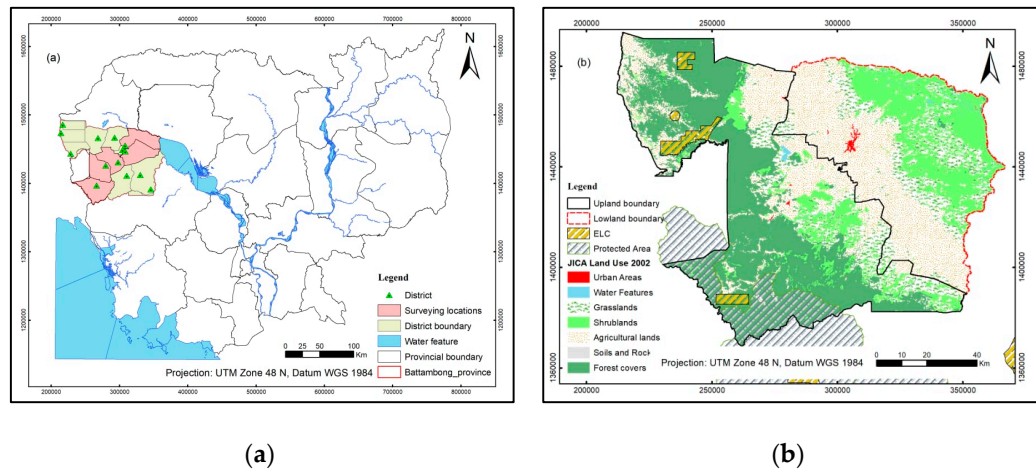

(**a**)          (**b**)

**Figure 1.** Map of the study area (**a**) Battambang province in Cambodia, and (**b**) protected areas (PAs) and economic land concession (ELC) in Battambang.

The province is divided into one municipality and 13 districts, with a total area of about 11,748 km$^2$. According to the General Population Census of Cambodia, the population of the province significantly grew from 793,129 in 1998 to 997,169 in 2018 [24]. The population in the uplands increased sharply, while the population in the lowlands decreased due to immigration (Table 1). In terms of population, it ranks fifth among the 24 provinces and one capital city, behind only Phnom Penh, Kandal, Prey Veng and Siem Reap [31]. Land cover types such as forest (evergreen and semi-evergreen), flooded forest, shrubland, water,

cropland and urban area are found throughout this northwestern province. Battambang is nationwide known as an agricultural hub. The agricultural production of the province has the largest area in the country. According to the 2019 Provincial Agricultural Report published by the Provincial Department of Agriculture, Forestry and Fisheries, paddy fields covered 699,944 hectares while other crops were grown on 297,312 hectares. Permanent crops such as mango (*Mangifera indica*), longan *(Dimocarpus longan)* and cashew (*Anacardium occidentale*) are mostly planted in the uplands. Not different from the national climate condition, Battambang is under the influence of tropical monsoon climate which consists of two main seasons: rainy season and dry season. Rainy season starts in May and ends in October, while dry season period is from November to April. The province has a yearly average temperature 27.7 °C and an annual rainfall of 1331 mm. However, the temperature and the average rainfall differed between upland and lowland (Table 1) [29,32].

**Table 1.** Annual rainfall, maximum (max) and minimum (min) temperature, elevation and population.

| Item | Annual Rainfall (mm) [29,32] | Temperature (°C) [32] | | Elevation (m) [30] | | | Population [23] | |
|---|---|---|---|---|---|---|---|---|
| | | Max | Min | Max | Average | Min | 1998 | 2018 |
| Low land | 1244 | 33.69 | 23.60 | 89 | 9.43 | 0 | 562,810 | 518,605 |
| Upland | 1419 | 32.26 | 23.03 | 1333 | 118.12 | 9 | 230,319 | 478,564 |

### 2.2. Satellite Images and Reference Data

The LULC change in Battambang province was analyzed based on Landsat data. Landsat images (Collection 2 Level-2) for five years (1998, 2003, 2008, 2013 and 2018) were downloaded from the United States Geological Survey website (https://earthexplorer.usgs.gov/ (accessed on 3 August 2019)). Landsat Collection 2 substantially improves the absolute geolocation accuracy of the global ground reference dataset, which enhances the interoperability of the Landsat archive through time. Collection 2 Level-2 includes surface reflectance and surface temperature data. In this study, we used only the surface reflectance product, which has been corrected for atmospheric effects of gases, aerosols, and water vapor. This correction is essential to reliably characterize the Earth's land surface. The spatial resolution of the images was 30 m. The downloaded images were from the dry season, which is the period in which the satellite can capture good quality images with no cloud cover or low percentage of cloud cover. For each year, two Landsat scenes (path/row: 127/51 and 128/51) were required in order to cover the whole study area (Table 2). The Landsat data were mosaicked and then clipped to the study area using the shapefile of Battambang provincial boundary.

**Table 2.** Landsat images used for LULC classification in this study of Battambang province.

| Year | Day and Month | Landsat Path/Row | Satellite/Sensor | Band |
|---|---|---|---|---|
| 1998 | 19 March | 128/51 | Landsat 5 TM | |
| | 29 April | 127/51 | | Band 1 to Band 7 (Blue, Green, Red, NIR, SWIR1, Thermal Infrared, SWIR 2) |
| 2003 | 04 January | 128/51 | Landsat 5 TM | |
| | 03 March | 127/51 | | |
| 2008 | 11 December | 127/51 | Landsat 5 TM | |
| | 20 December | 128/51 | | Band 1 to Band 11 (Coastal aerosol, Blue, Green, Red, NIR, SWIR 1, SWIR 2, Panchromatic, Thermal Infrared 1, Thermal infrared 2 |
| 2013 | 15 May | 127/51 | Landsat 8 OLI | |
| | 06 June | 128/51 | | |
| 2018 | 06 February | 128/51 | Landsat 8 OLI | |
| | 15 February | 127/51 | | |

In this study, reference data were collected from existing topographic and the land use map 1998 of the Geographic Department of Cambodia and land use maps produced by Japan International Cooperation Agency (JICA) in 2002; forest cover maps from 2002,

2006 and 2010 by the Ministry of Agriculture, Fishery and Forestry (MAFF), and historical images from Google Earth. In addition, field surveys using drone and handheld GPS were conducted in Battambang province in 2018. These reference data were used as training data for supervised classification and as validation data for accurate assessment of each LULC map.

### 2.3. Land Use and Land Cover Classification

A supervised classification technique using the maximum likelihood algorithm [33] was used to map LULC for each of the five years' images. The maximum likelihood algorithm assumes that the statistics for each class in each band are normally distributed, and calculates the probability that a given pixel belongs to a specific class. Each pixel is assigned to the class that has the highest probability (i.e., the maximum likelihood). Seven LULC classes, following the Cambodia land use map of 2002 produced by the JICA, were chosen for this study: urban/built-up area, water feature, grassland, shrubland, agricultural land, barren land, and forest cover. Urban/built-up area is characterized by high human population density and vast man-made structures such as residential and commercial areas, roads, and other construction. The surface of urban areas is highly sealed (e.g., buildings, roads). Water features consist of rivers, streams, lakes and man-made reservoirs. Grassland is land covered with grasses and other herbaceous species which might be used for pastures and grazing. There are many different types of grassland, designated by ecological zone, topography, climate and soil conditions. Shrubland is characterized by vegetation dominated by shrubs and bushes. Grasses, herbs, and geophytes may also be part of shrubland, which either occurs naturally or is the result of anthropogenic activities. Agricultural land corresponds to croplands which include paddy rice fields, annual crops, vegetables, fruit trees and orchards. Barren land is either uncovered or only slightly (not completely) covered soil and rocky ground. Forest cover consists of deciduous forest, coniferous and evergreen forest, and mixed forest.

In the process of LULC classification, Band 1 to 5 and Band 6 of the Landsat 5 TM, and Band 2 to Band 7 of the Landsat 8 OLI were used for the classification. The LULC classification was carried out using QGIS 3.10.

### 2.4. Accuracy Assessment

Accuracy assessment is the final and critical part of image classification process [34]. It is useful in the evaluation of classification techniques for determining the level of error that might be contributed by the image. The accuracy of each classification is expressed in the form of an error matrix [35–37]. It is defined as a square array of numbers in the columns and the rows illustrating the classes of information [34]. For the classified LULC maps of 1998, 2003, 2005, 2013 and 2018, there were, respectively, a total of 121, 430, 163, 308 and 317 validation points chosen randomly. We ensured that at least 10 validation points were included in each class. The validation points were derived from the existing maps of land use of 1998 from the Geographic Department and the land use map of 2002 from the JICA, the forest cover maps of 2002, 2006 and 2010 by MAFF, Google Earth images, drone images and handheld GPS data collected in 2018. Four common measures of classification accuracy derived from the error matrix were: overall accuracy, user's accuracy, producer's accuracy, and Kappa coefficient [36,38–40]. These measures were calculated as:

$$\text{Overall accuracy} = \frac{\sum_{i=1}^{r} n_{ii}}{n} \times 100$$
$$\text{User's accuracy} = \frac{n_{ii}}{n_{i+}}$$
$$\text{Producer's accuracy} = \frac{n_{ii}}{n_{+i}}$$
$$\text{Kappa coefficient} = \frac{n \sum_{i=i}^{r} n_{ii} - \sum_{i=1}^{r} (n_{i+} * n_{+i})}{n^2 - \sum_{i=1}^{r} (n_{i+} * n_{+i})}$$

where, $i$ is the class number; $n$ is the total number of classified pixels compared to reference pixels, $n_{ii}$ is the number of correctly classified pixels in each class, $n_i+$ is the total number

of classified pixels belonging to class *i*, and *n+i* is the total number of reference pixels belonging to class *i*.

### 2.5. Land Use and Land Cover Change Analysis

A post-classification comparison of LULC maps was implemented to determine LULC change over the 20-year period between 1998 and 2018. We also further analyzed changes for four five-year periods: 1998–2003, 2003–2008, 2008–2013 and 2013–2018. We used three main indices to statistically quantify the LULC change rate and trends [41–43]. The first index is defined as:

$$A_{ir} = (A_{it2} - A_{it1})/(t_2 - t_1)$$

where $A_{it1}$ and $A_{it2}$ denote the total area of the LULC type *i* at times $t_1$ and $t_2$; and $A_{ir}$ is the change rate per year for each LULC type from time $t_1$ to time $t_2$. The second index is calculated as:

$$A = \left[\sum_{i,j}^{n} \frac{dA_{i \rightarrow j}}{A_i}\right] \times (1/t) \times 100\%$$

where *Ai* is the total area of LULC type *i* in the study area at the initial year of the period, $dA_{i \rightarrow j}$ is the total area converted from LULC type *i* to LULC type *j*, t is the time period, A is the land use dynamic degree of the time period t, and *n* represents the LULC types. In order to determine the major LULC conversions, the contribution rate (ai%) of LULC type I is calculated as follows:

$$ai\% = (A_{j \rightarrow i} - A_{i \rightarrow j})/A_{ic} \times 100\%$$

where $A_{j \rightarrow i}$ is the conversion area of LULC type j to LULC type i, $A_{i \rightarrow j}$ is the conversion area of LULC type i to type j, and $A_{ic}$ is the change area of LULC type i in a certain period. The flow chart analysis of the study region is shown in Figure 2.

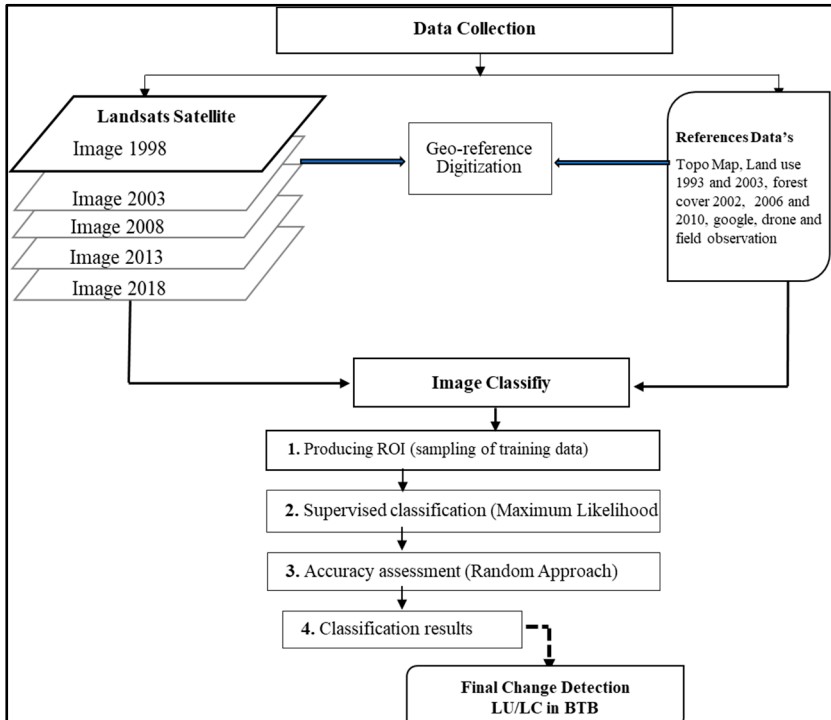

**Figure 2.** Flowchart of LULC change analysis in Battambang province.

## 3. Results and Discussion

### 3.1. Accuracies of LULC Classification

Multi-temporal Landsat images of 1998, 2003, 2008, 2013 and 2018 were classified, LULC maps were generated and accuracies were measured. Overall accuracies, user's accuracies, producer's accuracies and Kappa coefficients of the LULC maps (1998, 2003, 2008, 2013 and 2018) are summarized in Table 3. The overall accuracies of the maps were 93% for 1998, 82% for 2003, 94% for 2008, 93% for 2013 and 84% for 2018, providing sufficiently reliable data, which were referred to [44] to investigate LULC changes. The Kappa coefficients for all the years were greater than 0.77. Both user's and producer's accuracies for individual classes were very high for agricultural land, barren land, and forest cover, ranging from 38% to 100%.

**Table 3.** Land use and land cover classification accuracies.

| Year / LULC Classes | | 1 | 2 | 3 | 4 | 5 | 6 | 7 | Overall Accuracy (%) | Overall Kappa |
|---|---|---|---|---|---|---|---|---|---|---|
| 1998 | User (%) | 83 | 91 | 90 | 94 | 96 | 75 | 100 | 93 | 0.92 |
| | Producer (%) | 83 | 91 | 90 | 94 | 96 | 75 | 100 | | |
| 2003 | User (%) | 100 | 100 | 69 | 69 | 85 | 75 | 93 | 82 | 0.77 |
| | Producer (%) | 38 | 82 | 73 | 69 | 95 | 75 | 87 | | |
| 2008 | User (%) | 100 | 100 | 89 | 91 | 95 | 100 | 100 | 94 | 0.93 |
| | Producer (%) | 100 | 94 | 91 | 91 | 98 | 100 | 96 | | |
| 2013 | User (%) | 100 | 63 | 87 | 88 | 98 | 75 | 94 | 93 | 0.89 |
| | Producer (%) | 100 | 63 | 87 | 88 | 98 | 75 | 94 | | |
| 2018 | User (%) | 65 | 95 | 71 | 80 | 86 | 100 | 100 | 84 | 0.80 |
| | Producer (%) | 100 | 89 | 69 | 83 | 90 | 100 | 100 | | |

Note: 1-built-up area, 2-water feature, 3-grassland, 4-shrubland, 5-agricultural land, 6-barren land and 7-forest cover.

### 3.2. Land Use and Land Cover in Battambang Province

Figure 3 illustrates the spatial distribution of land use and land cover types in Battambang province for 1998, 2003, 2008, 2013 and 2018. Table 4 shows the areas and percentages of LULC classes in Battambang from 1998 to 2018.

Agricultural land makes up the largest percentage of Battambang province, with 44.5%, 39.2%, 61.1%, 68.9%, 68.4% in 1998, 2003, 2008, 2013 and 2018, respectively (Figure 3). The second largest LULC class was forest cover in 1998 and 2003, grassland in 2008, and shrubland in 2013 and 2018. Agricultural land was seen in both the flooded lowland areas and the upland part of the province, while forest occurred predominantly in the higher elevation area in the western part of the province. Shrubland was primarily found in the northeast part, where flooding from Tonle Sap Lake occurs annually. Built-up area, water feature, and barren land covered only a small portion of Battambang province, with these three LULC types together representing just below 1.6% in all the five years (Table 4).

### 3.3. Assessment of LULC Change at Different Time Periods

Between 1998 and 2018, the LULC changes in Battambang varied from one LULC class to another. The change of each LULC class at different time periods (1998–2003, 2003–2008, 2008–2013, 2013–2018 and 1998–2018) is presented in Table 5.

Forest cover had the largest area decrease of around 284,500 hectares (79%) between 1998 and 2018. The overall decrease rate was 14,200 hectares per year. The decrease rate of the forest area was fastest during 2003–2008 (42,200 hectares per year). However, it should be noted that the forest area slightly increased at a rate of 4300 hectares per year during 2008–2013. In contrast to forest cover, agricultural land experienced the largest increase in area (287,600 hectares or 54%) over this 20-year period, at a change rate of 14,300 hectares per year. Despite this general increase, there were two periods that saw drops in the area of agricultural land: 1998–2003 (63,700 hectares) and 2013–2018 (6000 hectares). The change of agricultural land during 2003–2008 was the greatest, with its area increasing

by 263,600 hectares. This number represents 91% of the total increase in agricultural land during the entire 20-year period, indicating that agricultural land expansion was intense during this short period of 2003–2008.

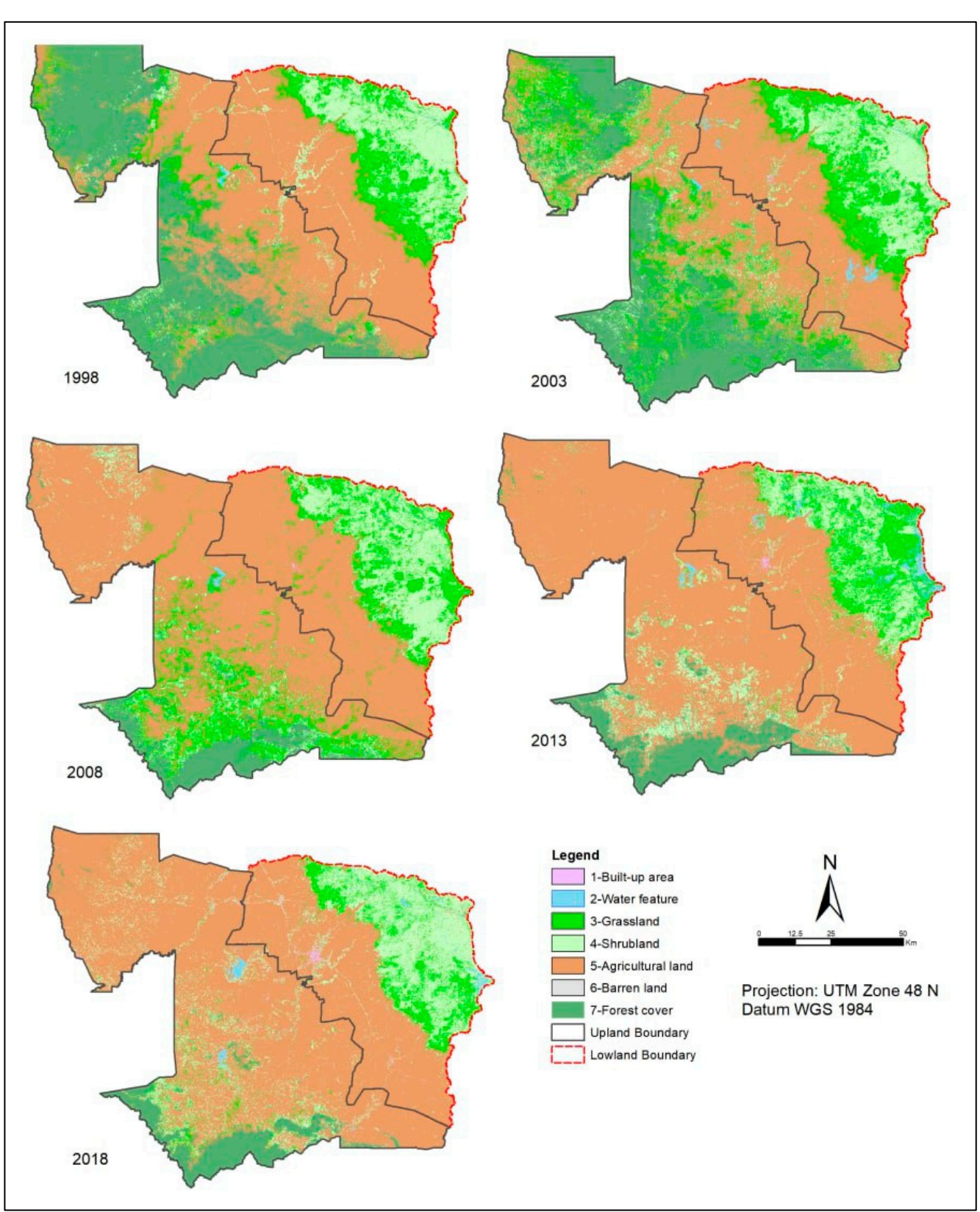

**Figure 3.** Spatial distributions of LULC in Battambang province in 1998, 2003, 2008, 2013 and 2018.

**Table 4.** Total areas and percentages of LULC classes in Battambang province, Cambodia (1998–2018).

| LULC Type | Area (1998) | | Area (2003) | | Area (2008) | | Area (2013) | | Area (2018) | |
|---|---|---|---|---|---|---|---|---|---|---|
| | Hectares | % | Hectares | % | Hectares | % | Hectares | % | Hectares | % |
| 1. Built-up area | 48 | 0.00 | 1106 | 0.09 | 302 | 0.03 | 1362 | 0.11 | 4698 | 0.39 |
| 2. Water feature | 2309 | 0.19 | 10,072 | 0.84 | 4707 | 0.39 | 17,404 | 1.45 | 10,637 | 0.88 |
| 3. Grassland | 151,753 | 12.61 | 253,106 | 21.03 | 208,051 | 17.29 | 99,793 | 8.29 | 75,683 | 6.29 |
| 4. Shrubland | 154,916 | 12.87 | 179,305 | 14.90 | 178,717 | 14.85 | 158,083 | 13.13 | 213,575 | 17.74 |
| 5. Agricultural land | 535,627 | 44.50 | 471,904 | 39.21 | 735,584 | 61.11 | 829,278 | 68.90 | 823,225 | 68.40 |
| 6. Barren land | 16 | 0.00 | 995 | 0.08 | 224 | 0.02 | 45 | 0.00 | 1395 | 0.12 |
| 7. Forest cover | 358,960 | 29.82 | 287,140 | 23.86 | 76,042 | 6.32 | 97,664 | 8.11 | 74,416 | 6.18 |
| Grand Total | 1,203,628 | | 1,203,628 | | 1,203,628 | | 1,203,628 | | 1,203,628 | |

**Table 5.** Total area and percentage of LULC change between the years 1998–2003, 2003–2008, 2008–2013, 2013–2018 and 1998–2018.

| LULC Type | 1998–2003 Net Area Change | | 2003–2008 Net Area Change | | 2008–2013 Net Area Change | | 2013–2018 Net Area Change | | 1998–2018 Net Area Change | |
|---|---|---|---|---|---|---|---|---|---|---|
| | Hectares | % | Hectares | % | Hectares | % | Hectares | % | Hectares | % |
| 1. Built-up area | 1058 | 2196 | −804 | −73 | 1060 | 351 | 3336 | 245 | 4650 | 9651 |
| 2. Water feature | 7763 | 336 | −5365 | −53 | 12,696 | 270 | −6766 | −39 | 8328 | 361 |
| 3. Grassland | 101,352 | 67 | −45,055 | −18 | −108,267 | −52 | −24,102 | −24 | −76,071 | −50 |
| 4. Shrubland | 24,389 | 16 | −588 | 0 | −20,648 | −12 | 55,505 | 35 | 58,659 | 38 |
| 5. Agricultural land | −63,723 | −12 | 263,680 | 56 | 93,620 | 13 | −5979 | −1 | 287,598 | 54 |
| 6. Barren land | 979 | 6225 | −771 | −77 | −180 | −80 | 1350 | 3001 | 1379 | 8766 |
| 7. Forest cover | −71,820 | −20 | −211,098 | −74 | 21,613 | 28 | −23,239 | −24 | −284,543 | −79 |

Similarly, shrubland also experienced a noticeable change during the study period. Its area increased by 58,600 hectares from 154,900 hectares (12.87% of the study area) in 1998 to 213,600 hectares (17.74%) in 2018 (Table 4). As can be seen in Table 5, this increase mostly occurred during the last five years (2013–2018). Almost no change of shrubland area was seen during 2003–2008. For the grassland area, there was a substantial increase during 1998–2008, and then decreases for the other periods. Overall, the grassland area fell by 76,000 hectares, at an annual rate of 3800 hectares (Table 5).

The built-up area grew by 465,000 hectares between 1998 and 2018. Although the change of this LULC type was very small in terms of area, the relative change was the highest among all the LULC types (nearly 1000%). Such immense change reflects the rising population and urbanization in the province since the end of civil war. Noticeably, the built-up area increased more rapidly during the past 10 years (2008–2018). For water features, their area increased by 8300 hectares over this 20-year period. The increase was mainly due to the construction of the Battambang Multipurpose Dam located in Ratanak Mondul district.

Figure 4 illustrates the spatial distribution of the LULC changes for the different time periods. The change maps provide information on the transition of each LULC type to the other classes, revealing noticeable LULC dynamics not only for the entire 20-year period, but also for the four 5-year periods. More precisely, the first two periods (1998–2003 and 2003–2008) saw greater conversions than the last two periods (2008–2013 and 2013–2018). For the entire study period, stable areas were mostly located in the province's middle part along the national road 5. These areas have been traditionally used as farmland and human settlements for many years. Stable forest cover was found in mountainous areas which were designated as naturally protected areas, such as the Phnom Samkos Wildlife Sanctuary and the Samlout Multiple Use Area. Between 1998 and 2018, conversion of forest land to other LULC types was most obvious compared to other individual types, taking place significantly in upland areas in the northwest and west portions of the province. Conversions of shrubland and grassland to the other LULC

types were apparent too, especially in lowland areas near Tonle Sap Lake. According to Table 5, the amount of forest extent loss and agricultural land expansion between 1998 and 2018 were comparable (284,500 hectares vs. 287,500 hectares). Thus, it is probable that forest area was lost to agricultural land. However, other LULC types would also contribute to increased agricultural land, as seen in the LULC change detection matrices (Tables A1–A5). Generally, for a certain LULC type there are three kinds of change: no conversion, conversion to other LULC types, and conversion from other LULC types. These data are provided in the Appendix A to better understand the change of each LULC type.

The LULC change detection matrices for different time periods (Tables A1–A5) show the conversion from each class to other individual classes. For instance, considering the entire study period of 1998–2018 (Table A5), 499,900 hectares of agricultural land remained unchanged, 323,100 hectares of new agricultural land were created from conversions of forest cover (233,200 hectares), grassland (60,700 hectares), shrubland (28,700 hectares) and the other categories (400 hectares). Moreover, 35,600 hectares of agricultural land were lost from conversions to shrubland (23,700 hectares), grassland (4700 hectares), built-up area (3000 hectares), water features (1900 hectares), forest cover (1400 hectares) and barren land (900 hectares). Agricultural land was the LULC type that expanded the most (323,100 hectares), followed by shrubland (122,200 hectares) and grassland (44,200 hectares). At the same time, the LULC types that had the highest loss were forest cover, shrubland and grassland, with the loss areas of 289,800 hectares, 120,200 hectares and 63,600 hectares, respectively.

### 3.4. Analysis of Major LULC Type Conversion

Since agricultural land and forest cover were the top two LULC types with respect to the overall change, they were chosen for further analysis. Table 6 shows that agricultural land was mainly changed to grassland and shrubland, accounting for 69.7% and 25.4% of the change area between 1998 and 2003, respectively. Built-up area, barren land and water features slightly contributed to decrease of the agricultural land. The decline of agricultural land during this five-year period was because the land was abandoned or left uncultivated, allowing grass or small trees to grow. However, after this period, agricultural land did not continue to convert to grassland or shrubland, and was reversely converted from these LULC types. For the period of 2003–2008, the increase of agricultural land was largely offset by the conversion of grassland with a contribution rate of around 39%, while shrubland contributed to an increase in agricultural land to some extent. During this period, forest area was also converted to agricultural land at virtually the same rate as grassland. Furthermore, as shown in Table 6, the changes in the area of agricultural land in relation to forest cover were positive for all periods, which means that forest cover had been continuously converted to agricultural land. Therefore, it is clear that the main source of the increase of agricultural land was conversion from forest cover.

In addition, Table 6 shows that there were decreasing trends of forest cover in all periods, except for 2008–2013. During 1998–2003, grassland had the highest contribution rate (about 72%), followed by shrubland (22.3%) and agricultural land (4.5%). However, the forest area during this period did not change as much as the next period. During the period of 2003–2008, there were many changes between forest cover and other three LULC types (i.e., agricultural land, grassland and shrubland). Agricultural land was the largest contributor to the change, accounting for nearly half of the total change area of forest cover during this period. During 2013–2018, agricultural land remained the highest contributor (52.7%) although the net change of forest area was relatively small. Water features, barren land and built-up areas only slightly contributed to the forest cover decrease.

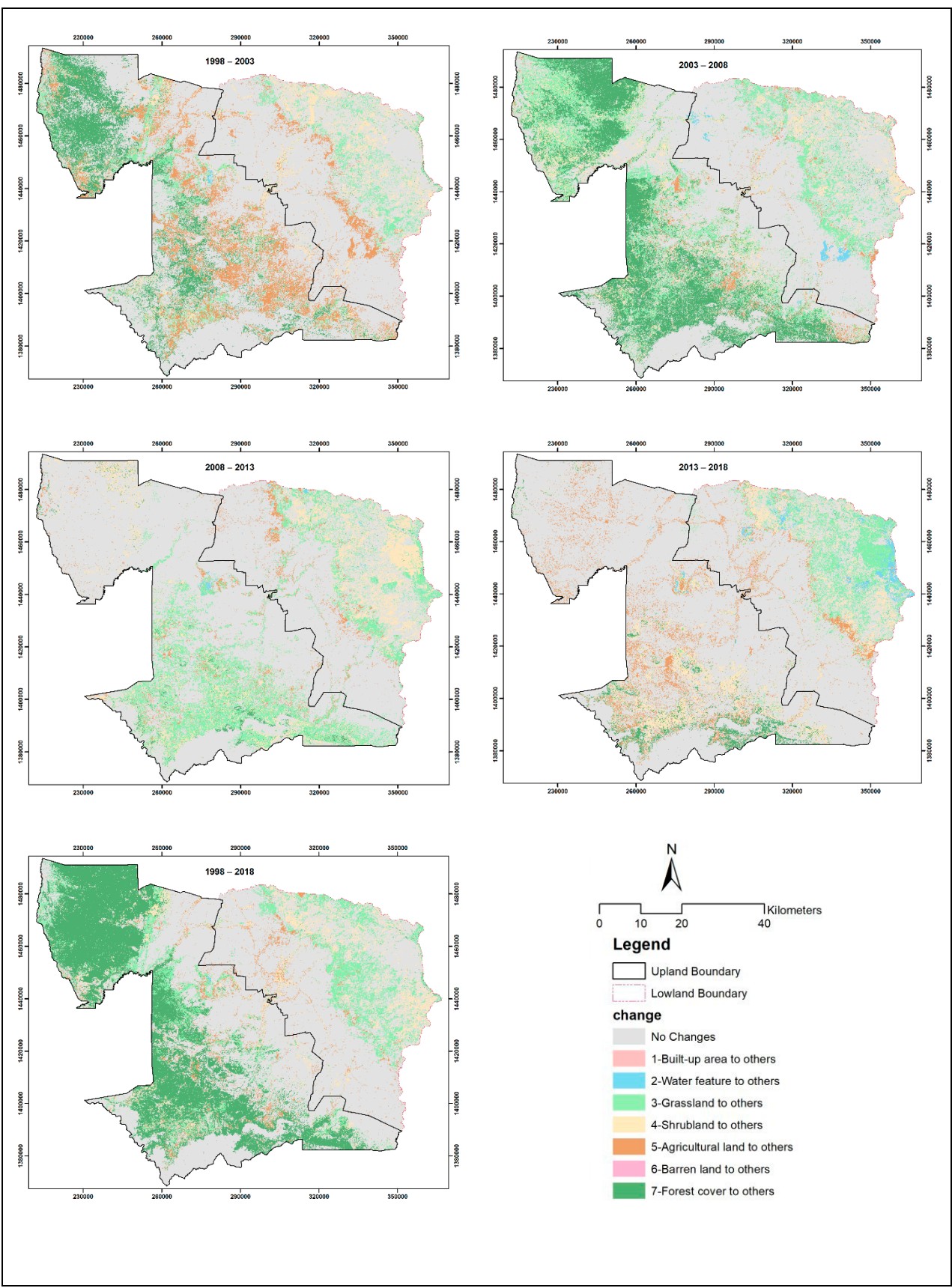

**Figure 4.** Spatial distribution of the LULC change in Battambang province for the different time periods.

**Table 6.** Conversion and contribution rates of other LULC types for forest cover and agricultural land in different periods.

| LULC Type Change | 1998–2003 | | 2003–2008 | | 2008–2013 | | 2013–2018 | | 1998–2018 | |
|---|---|---|---|---|---|---|---|---|---|---|
| | NC (ha) | CR (%) | NC (ha) | CR (%) | NC (ha) | CR (%) | NC (ha) | CR (%) | NC (ha) | CR (%) |
| AL-FC | 3207 | −5.0% | 103,423 | 39.2% | 6473 | 6.9% | 12,236 | −204.6% | 231,846 | 80.6% |
| AL-SL | −16,203 | 25.4% | 50,909 | 19.3% | 12,793 | 13.7% | −10,316 | 172.5% | 5027 | 1.7% |
| AL-GL | −44,391 | 69.7% | 103,920 | 39.4% | 75,705 | 80.9% | −2906 | 48.6% | 56,013 | 19.5% |
| AL-BL | −831 | 1.3% | 666 | 0.3% | 186 | 0.2% | −1259 | 21.1% | −857 | −0.3% |
| AL-BA | −647 | 1.0% | 743 | 0.3% | −1055 | −1.1% | −3154 | 52.8% | −2995 | −1.0% |
| AL-WF | −4847 | 7.6% | 4302 | 1.6% | −459 | −0.5% | −662 | 11.1% | −1517 | −0.5% |
| FC-AL | −32.07 | 4.5% | −103,423 | 49.0% | −6473 | −29.9% | −12,236 | 52.7% | −231,846 | 81.5% |
| FC-SL | −16,014 | 22.3% | −42,885 | 20.3% | 11,713 | 54.2% | −7020 | 30.2% | −42,060 | 14.8% |
| FC-GL | −51,660 | 71.9% | −64,604 | 30.6% | 16,436 | 76.0% | −3882 | 16.7% | −8390 | 2.9% |
| FC-BL | −82 | 0.1% | −31 | 0.0% | 0.00 | 0.0% | −6 | 0.0% | −352 | 0.1% |
| FC-BA | −164 | 0.2% | −9 | 0.0% | 19 | 0.1% | −28 | 0.1% | −357 | 0.1% |
| FC-WF | −589 | 0.8% | −2 | 0.0% | −50 | −0.2% | −62 | 0.3% | −1356 | 0.5% |

Note: BA = built-up area, WF = water feature, GL = grassland, SL = shrubland, AL = agricultural land, BL = barren land and FC = forest cover; NC = net change; CR = contribution rate; ha = hectare.

### 3.5. Comparison between LULC Change in Upland Area and in Lowland Area

The upland area and lowland area of Battambang province were separated based on elevation and the district boundary. The districts next to Tonle Sap Lake are defined as lowland. The areas of LULC types in the upland and lowland areas of Battambang province for various time periods are shown in Figure 5. There were more forests in the upland area than in the lowland area for all years. Forest cover makes up only a small portion of the lowland area, and its area did not change much over the years. However, forest cover in the upland area saw a general decrease, with the largest drop taking place between 2003 and 2008 (Figure 5). The upland forest area decreased from 355,500 hectares in 1998 to 279,300 hectares in 2003, then to 74,600 hectares in 2008. From 2008, the area only changed slightly until 2018. Agricultural land in the lowland area did not experience obvious change, but in th upland area it increased dramatically, particularly between 2003 and 2008. Between 1998 and 2000, the area of agricultural land in the upland area slightly dropped to 241,800 hectares, but the area doubled to 483,700 hectares in 2008. It continued to increase by 92,100 hectares in 2013 and then slightly dropped from 575,800 hectares to 572,700 hectares in 2018. For grassland and shrubland, there were fluctuations in their areas in both the uplands and the lowlands, but there was a general decrease. Built-up area in both the uplands and lowlands showed a rapid increase until 2018, with the exception of 2008. For the entire study period of 1998–2018, forest cover and grassland decreased in both the uplands and lowlands, but the levels of change in uplands were higher than those of lowlands. The remaining LULC types experienced increases. The areas of these LULC types increased more in the uplands than in the lowlands, with the exception of water features whose area grew more in the lowlands than in the uplands.

### 3.6. Main Drivers of Land Use and Land Cover Change in Battambang Province

3.6.1. Expansion of Agricultural Land and Reduction of Forest Cover

The results of LULC change analysis showed that between 1998 and 2018 agricultural land increased dramatically at the expense of forest cover and grassland, and the transition from these LULC types to agricultural land occurred mainly during the period of 2003–2008.

There were several policies, legal frameworks and projects which influenced LULC (Table 7). In late 1998, the win-win policy was declared to end the Cambodian civil war, particularly in the northern Cambodia, including Battambang province. According to interviews with local authorities, following the ceasefire some families moved to their homeland to find their relatives leaving their farmland unused. Through natural processes, the land may become grass or shrubs. This was the reason why agricultural land declined by 12% during the period 1998–2003.

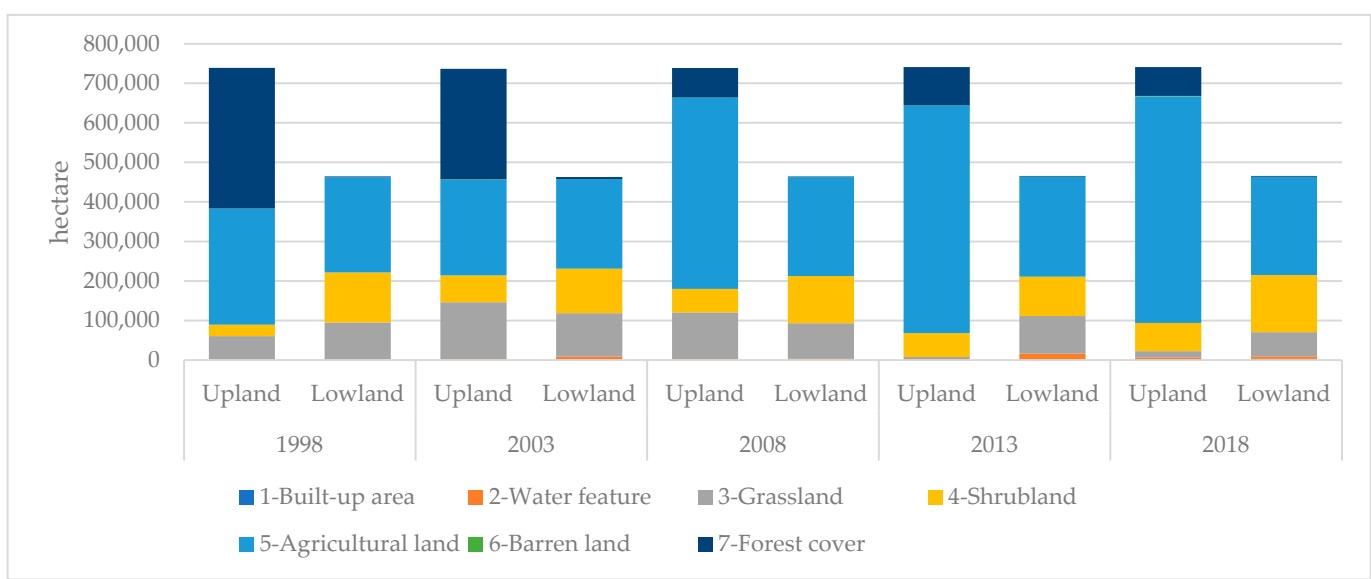

**Figure 5.** Detailed dynamics of LULC change in the upland and lowland areas of Battambang between 1998 and 2018.

**Table 7.** Policies, legal framework and projects affecting LULC change in Battambang province between 1998 and 2018.

| Year | Category | Content | Level |
|---|---|---|---|
| 1998 | Policy | Win-Win Policy ending the civil war and peacefully reintegrating the Khmer Rouge | National |
| 1996 | Law | Law on Environmental Protection and Natural Resource Management | National |
| 2001 | Law | Land Law 2001 | National |
| 2002 | Law | Forestry Law 2002 | National |
| 2003 | Regulation | Sub-decree on Social Land Concession | National |
| 2004 | Regulation | Sub-decree on Socio-Economic Management of Mine Clearance Operations | National |
| 2005 | Regulation | Sub-decree on Economic Land Concession | National |
| 2007 | Regulation | Circular on Measures Against Illegal Holding of State Land | National |
| 2012 | Regulation | Order 01BB on the Measures Strengthening and Increasing the Effectiveness of the Management of Economic Land Concessions (ELC) | National |
| 2012–2013 | Project | Implementation of Order 01 dated 07 May 2012 on Measures Strengthening and Increasing Effectiveness of Economic Land Concession Management | National |

According to Rasmussen [45], the northwest part of Cambodia that was ruled by Khmer Rouge was contaminated by landmines. Landmine clearance work fully started in 1992. To ensure the landmine clearance was in line with planning, in 2004 the Cambodian government issued Sub-decree No. 70 on Socio-Economic Management of Mine Clearance Operations, which outlined the role of key mine action actors (e.g., Cambodian Mine Action and the Victim Assistance Authority) [46]. Between 1992 and 2016, approximately 154,500 hectares of minefields throughout Cambodia were released for use as cultivated land [47]. In Battambang, the released minefields were not only given to the returning families of Khmer Rouge soldiers, but also granted to poor and landless families to build houses and cultivate crops [25]. Land Law 2001 allows two legal mechanisms to transfer state-owned land (e.g., forest land) namely Social Land Concession (SLC) and Economic Land Concession (ELC). SLC is to transfer private state land for social purposes to the poor who lack land for residential and/or family farming purposes, while ELC grants private state land through a specific economic land concession contract to a concessionaire to use for agricultural and industrial agricultural purposes. The government issued the sub-decree on SLC and sub-decree on ELC in 2003 and 2005, respectively. According to interviews with forest officials and land officials, 21,480 hectares of forest land have been

granted to private companies through ELC for agricultural purposes, and 1026 hectares in Samlout district were converted from forest land in order to offer to landless families through social land concession from 2010 to 2011 in Battambang province. The population of Battambang province reached over one million in 2008, which resulted in increased demands for agricultural land. In addition, some encroachments of forest land have also been reported. In 2012, the Cambodian Government issued Order 01 BB on the Measures Strengthening and Increasing the Effectiveness of the Management of ELCs to reinforce and increase the efficiency of land management with an emphasis on reducing land conflicts and providing titles to incumbent landholders. Following this Order, the Government employed a large-scale workforce to implement systematic issuance of private land titles for 1.2 million hectares of land covering 350,000 families living within ELC areas and state-owned forest land [48,49]. All these aforementioned policies, laws and factors prompted the large increase in agricultural land of 263,600 hectares between 2003 and 2008, the moderate increase of 13% (93,600 hectares) during 2008–2013, and the decrease in forest cover of 282,900 hectares between 1998 and 2013.

According to [50–52], cultivated land was expanded leading to increases of pastures in Mexico and soybeans in South America, in accordance with the agricultural expansion observed in study in area for cassava, corn and other fruit trees in the uplands. The goal of the Cambodian agricultural sector development strategy plan (2019–2023) is to increase all type of agricultural production around 10% per year [53]. However, there were no clear estimates on how much area of natural land will be converted to agricultural land in the future. Such data is important to develop, and suggestions and recommendations for decision makers to develop sustainable, inclusive and resilient land use plans are needed. Our research methodology and results fill this gap and provides valuable data.

During 2013–2018, agricultural land slightly decreased, mainly due to the regrowth of shrubland. The population of Battambang fell slightly from over 1 million in 2008 to just below 1 million in 2018 as some people went abroad as migrant workers. According to Cambodia's Ministry of Agriculture, Forestry and Fisheries, the total labor force decreased from 57.6% in 2009 to 45.3% in 2014. These factors may potentially be the cause of this minor drop in agricultural land area.

Forest cover was also significantly converted to shrubland (42,000 hectares) during the entire study period of 1998–2018 (Table A5). Illegal logging and deforestation could partly contribute to the change because this activity degraded the forest, which subsequently evolved to shrubland. The change was also a result of the ineffective implementation of Cambodian Law on Environmental Protection and Natural Resource Management, and Forestry Law 2002 [54]. The findings in Kampong Thom of Cambodia [55] and from Indonesia [56,57] also revealed that the decline of forest area was caused by illegal logging and weak enforcement of policies.

### 3.6.2. Increase in Built-Up Area

This study revealed that 4650 hectares (9651%) of built-up area increased in the period of 1998–2018, and it was mainly converted from agricultural land and shrubland. The increase of the built-up area in Battambang province was related to the population growth, socio-economic development, and infrastructure development. The large-scale increase of build-up area was observed in other regions around the world, as observed in Malagarasi, Uasin Gishu, Trans-Nzoia and Tanzania in Africa as a result of conversion from forest cover, grassland and wetland [58–60].

According to National Institute of Statistics, the population in Battambang province had increased from 793,129 in 1998 to 1,025,174 persons in 2008 but decreased to 997,169 in 2018 (this figure did not include migrants working abroad of 178,401 people). The annual population growth rate of 2.3% during the last two decades was higher than the population growth rate for the whole country (1.4%). Migrants sent money to their families to spend on housing and to improve the economic situation of households [61]. This finding was in line with the other studies [60,62].

The enlarged built-up area in Battambang was also reflected in the improvement and development of infrastructures (Figure 6). Between 2001 and 2011, three categories of road (asphalt, laterite and dirt road) increased rapidly from 544 km to 2529 km in the north-western uplands of Battambang [20]. Between 2013 and 2018, bituminous, DBST (double bituminous surface treatment) and concrete roads were developed while constructed earth roads were not improved, according to the Battambang Planning Department [63].

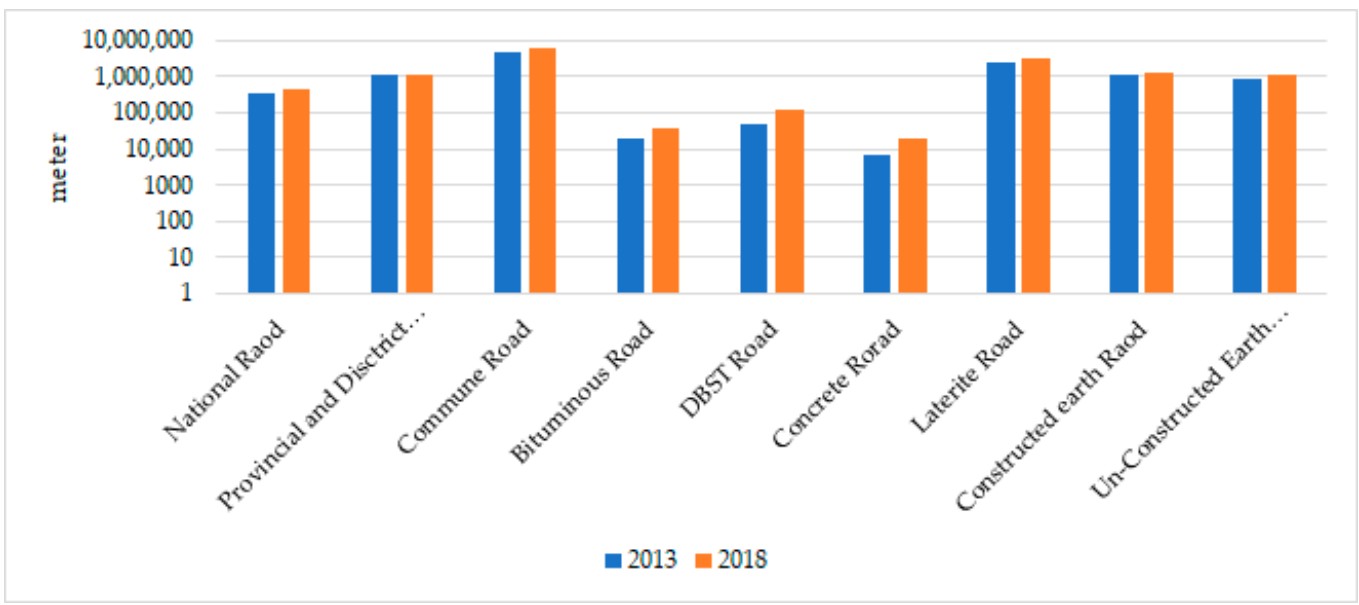

**Figure 6.** Type of roads in Battambang province in 2013 and 2018 [63].

### 3.6.3. Economic Growth and Rising Land Prices as a Driver of LULC Change

As economic conditions improved, land prices in Battambang province increased steadily. The prices differ from one land use type to another. Land prices in the capital city of Battambang in 2014 increased around 12% from 2013. According to key informant interviews during the field survey, the agricultural land price sharply increased from $650 to $20,000 US dollars per hectare over the past 20 years. The rising land price was found to be the driver of LULC change in the northeastern province of Rattanakiri [64]. In Battambang, it could potentially lead to conversion of agricultural land to urban built-up land, which has higher price as buyers or investors want to make profits. This was clearly evident in the increasing number of large residential housing development projects in the province in the past several years, with up to 10 projects as of 2017. The growing land price and market might also cause encroachments of public lands such as forest land, grassland and shrubland. Encroachers, mostly from other provinces, clear the vegetation and convert the land to agricultural land for sale. Kong et al. [20] reported that due to the growth of land price, farmers' plots were sold out, and then they explored new farming locations where forest cover or shrubland were present. This also happened in other countries, e.g., farmers living in the southern Amazon sold their agricultural land, and then encroached more forest land [14,60].

During the last 2 decades since 1998, Cambodia's gross domestic product (GDP) grew on average 7.87% per year [65]. According to Asian Development Bank (ADB) [66], the significant economic growth helped to move Cambodia from low-income country to lower middle-income country and contributed greatly to poverty reduction. However, this evolution also had negative consequence on forest cover. For instance, Cambodia retained only 8.51 million hectares (46.86%) of forest cover including rubber, palm oil, cashew nut and other fruit trees in 2018 [67]. Trisurat et al. [68] claimed that such large scale LULC changes that help economic growth also negatively impacted biodiversity in Thailand.

### 3.6.4. Climate and Environmental Drivers of Land Use and Land Cover Change

The climate and environmental change impacts negatively in Cambodia [69]. Flood, drought and storm are the three major climate-related disaster happening every year in Cambodia [70,71]. There is no exception for Battambang province. For example, 93,082 hectares and 27,340 hectares of agricultural land were damaged by flood and drought, respectively, in 2018 [72]. The flood mostly happened in the uplands due to storm water and deforestation, while the lowland in the particular area next to Tonle Sap experienced small floods due to deforestation and the power dam constructed along the Mekong River [73]. In 2020, 66,088 households and 1,188,703 meters of road were affected by severe flooding, and 4592 households were evacuated to the safe sites in Battambang province. Moreover, 164,116 hectares of agricultural land flooded in Battambang [74]. Furthermore, 70% of residents living in Koh Chivang in the lowlands of Battambang are fishermen. But, due to small floods, most of them have now shifted to agricultural farming on the land where they have cleared flooded forests [73].

According to focus group discussion (FGD) and key informant interview (KII), soil in upland areas has become less fertile and chemical fertilizer consumption has significantly increased in recent years, while the chemical fertilizer consumption in the lowland area did not increase much. This finding was also confirmed in the study of Kong et al. [20] and Touch et al. [75] which reported that the farmers have increased fertilizer utility to get high yields due to soil fertility decline.

These frequent natural disasters could also cause change in LULC. According to [20], environmental change encouraged the use of new cropping practices and it will become a driver of LULC change as well.

### 3.7. National and Global Relevance and Learning

The results from this study provided granular and high-resolution data on LULC in one of the key provinces of Cambodia where there were significant changes in deforestation and expansion in agricultural land and the overarching reasons for these activities. These data provide knowledge that will be valuable at the national level to implement policies to ensure that such changes do not happen in other provinces and regions in Cambodia. Such data will help find ways to better implementat and monitor policies in Cambodia, neighboring countries and around the world. Some of these regions include the Lower Mekong Basin, the 3S Basin (Srepok, Sesan, Sekong Rivers), Tonle Sap Lake, and in countries such as Myanmar, Laos, Thailand and Vietnam. These regions have shown accelerating deforestation, and appropriate monitoring and land use planning is required [28].

At the global level, deforestation and agricultural expansion continue to occur [76] including regions in Southeast Asia [77] and many tropical regions of Latin America, Africa and Asia [55,78,79]. Several countries in these regions are invested in developing national deforestation monitoring plans using different methods [76]. Therefore, crossing learning among different regions and data sharing provide opportunities to adapt and customize global data for effective monitoring, reporting and help with long-term sustainability [76]. Thus, the insights learned in our study will be valuable to other regions with similar biophysical and socioeconomic conditions.

### 4. Summary and Conclusions

Based on the analysis of Landsat satellite images in the study area, the results of this study revealed significant LULC change since the end of the civil war and Khmer Rouge integration from 1998 to 2018. The amounts of change in the five-year periods of 1998–2003, 2003–2008, 2008–2013 and 2013–2018 were different. The human footprint and population increased while natural cover degraded or was lost. Agricultural land and forest cover were the LULC types that experienced the largest changes between 1998 and 2018, with the area of agricultural land increasing by 287,600 hectares and the area of forest cover decreasing by 284,500 hectares. The majority of the change took place during the period of 2003–2008. Conversion analysis showed that forest cover was largely lost to

agricultural land and shrubland, while agricultural land was expanded mainly from forest cover, grassland and shrubland. The built-up area, an important human footprint, grew by 4600 hectares between 1998 and 2018. Noticeably, the built-up area increased more rapidly during the past 10 years. Despite its small area, it changed very fast (nearly 1000%). Such immense change reflects the rising population and urbanization in the province since the end of civil war.

The spatial change in LULC was affected by the terrain. Agricultural land mainly expanded in the upland area, and most of the expansion was through conversion from forest cover, grassland and shrubland. However, there was no noticeable change in agricultural land in the lowlands. The area of water feature increased in both upland land and lowland as a result of irrigation development. Overall, the levels of change were higher in upland areas, except for water feature and built-up area.

A number of main drivers of LULC change were identified over the period of study of 1998–2018. These drivers included policies, legal frameworks and projects to promote economic conditions, population growth, infrastructure development, economic growth and rising land prices, and climate and environmental change. The large increase of agricultural land was driven by the landmine clearance projects, social and economic land concessions and population growth, leading to more demands for cultivated land. This increase of agricultural land was primarily from the conversion of forest cover. However, forest cover was also significantly converted to shrubland. Illegal logging and deforestation were the factors that contributed to the change from forest to shrubland because this activity degraded forests, which subsequently evolved into shrubland. Population and economic growth not only resulted in an increase of built-up area, but also led to rising demand for agricultural land and rising land prices, which triggered the changes of other LULC types as well.

The LULC maps and change analysis explained the impact of past policies, legal framework, projects, and other drivers such as population trends, socio-economic conditions and climate/environmental change. Understanding of the main drivers of LULC change is important for predicting future LULC patterns and for establishing provincial spatial planning that is sustainable, inclusive and resilient. In addition, the result of driving forces of LULC in Battambang clearly shows that without any proper integrated forest resource management to balance forest conversion to agriculture and built-up areas, LULC changes can lead to the future weakening of the ecosystem services provided by forest biodiversity. The results of our study in the Battambang province of Cambodia serve as case study to learn and emphasize the need for the proper implementation of policies. Our data are useful for developing appropriate monitoring plans and strategies at national level in Cambodia and in global communities and regions where such efforts are needed. Therefore, further study regarding land policy management, soil degradation causing by LULC, economic costs-avoided from forest conversion to agriculture should be the conducted to develop holistic solutions for Cambodia and other countries facing similar problems.

**Author Contributions:** Conceptualization, T.S., S.P., N.N.; methodology, T.S., S.P., N.N., P.R.; software, T.S., P.R.; formal analysis, T.S., S.P., N.N.; investigation, P.C., D.T.; resources, T.S., N.N., P.V.V.P., M.R.R.; data curation, T.S.; writing—original draft preparation, T.S., S.P., N.N.; writing—review and editing, T.S., N.N., S.P., P.V.V.P., M.R.R., P.C., D.T.; visualization, T.S., N.N; supervision, S.P., P.C., D.T.; funding acquisition, P.V.V.P., M.R.R. All authors have read and agreed to the published version of the manuscript.

**Funding:** This research is made possible by the generous support of the American People provided to the Center of Excellence on Sustainable Agricultural Intensification and Nutrition (CE SAIN), the Royal University of Agriculture (RUA) and to the project on "Pattern and Drivers of Land use Change in Battambang Province" implemented by the Faculty of Land Management and Land Administration, RUA, through the Geospatial and Farming Systems Research Consortium (GFSRC), University of California, Davis (United States) under bilateral agreement No. 201403286-06 between the two universities Subaward #S15115. Funds to CE SAIN and GFSRC were provided for research and scholarship through the Feed the Future Innovation Lab for Collaborative Research on Sustainable

Intensification at Kansas State University, funded by the United States Agency for International Development (USAID) under Cooperative Agreement No. AID-OAA-L-14-00006. The contents are the sole responsibility of the authors and do not necessarily reflect the views of USAID or the United States Government or representing organizations.

**Institutional Review Board Statement:** Not applicable.

**Informed Consent Statement:** Not applicable.

**Data Availability Statement:** Data is available upon request from the corresponding author.

**Acknowledgments:** The authors thank Sanara Hor (Royal University of Agriculture, Cambodia), Robert J. Hijmans, Aniruddha Ghosh and Alex Mandel (University of California, Davis) for their support and guidance to conduct this research. Contribution number 22-098-J from Kansas Agricultural Experiment Station.

**Conflicts of Interest:** The authors declare no conflict of interest.

# Appendix A

**Table A1.** Land use and land cover (LULC) change detection matrix in hectare (ha) and percentage for 1998–2003.

| | LULC Class | LULC 2003 | | | | | | | | | | | | | | Grand Total | Loss |
|---|---|---|---|---|---|---|---|---|---|---|---|---|---|---|---|---|---|
| | | 1 | | 2 | | 3 | | 4 | | 5 | | 6 | | 7 | | | |
| | | ha | % | ha | % | ha | % | ha | % | ha | % | ha | % | ha | % | | |
| LULC 1998 | 1-Built-up area | 18 | 2 | 0 | 0 | 1 | 0 | 0 | 0 | 28 | 0 | 0 | 0 | 1 | 0% | 48 | 30 |
| | 2-Water feature | 4 | 0 | 1367 | 14 | 546 | 0 | 61 | 0 | 270 | 0 | 0 | 0 | 55 | 0% | 2303 | 936 |
| | 3-Grassland | 44 | 4 | 1775 | 18 | 75,081 | 30 | 31,967 | 18 | 22,551 | 5 | 45 | 5 | 20,209 | 7% | 151,672 | 76,591 |
| | 4-Shrubland | 198 | 18 | 1053 | 11 | 38,577 | 15 | 89,630 | 50 | 12,099 | 3 | 23 | 2 | 13,285 | 5% | 154,865 | 65,235 |
| | 5-Agricultural land | 675 | 61 | 5117 | 51 | 66,942 | 26 | 28,302 | 16 | 407,997 | 86 | 843 | 85 | 25,658 | 9% | 535,533 | 127,537 |
| | 6-Barren land | 0 | 0 | 0 | 0 | 2 | 0 | 0 | 0 | 12 | 0 | 0 | 0 | 1 | 0% | 16 | 15 |
| | 7-Forest cover | 165 | 15 | 644 | 6 | 71,869 | 28 | 29,299 | 16 | 28,865 | 6 | 83 | 8 | 227,739 | 79% | 358,665 | 130,926 |
| | **Grand Total** | 1103 | | 9956 | | 253,018 | | 179,260 | | 471,823 | | 995 | | 286,947 | | 1,203,103 | |
| | **Expansion** | 1085 | | 8589 | | 177,937 | | 89,630 | | 63,826 | | 995 | | 59,208 | | | |

**Table A2.** Land use and land cover (LULC) change detection matrix in hectare (ha) and percentage for 2003–2008.

| | LULC Class | LULC 2008 | | | | | | | | | | | | | | Grand Total | Loss |
|---|---|---|---|---|---|---|---|---|---|---|---|---|---|---|---|---|---|
| | | 1 | | 2 | | 3 | | 4 | | 5 | | 6 | | 7 | | | |
| | | ha | % | ha | % | ha | % | ha | % | ha | % | ha | % | ha | % | | |
| LULC 2003 | 1-Built-up area | 118 | 39 | 5 | 0 | 84 | 0 | 7 | 0 | 887 | 0 | 1 | 0 | 3 | 0 | 1104 | 986 |
| | 2-Water feature | 0 | 0 | 1896 | 41 | 2277 | 0 | 329 | 0 | 5429 | 1 | 0 | 0 | 74 | 0 | 10,004 | 8108 |
| | 3-Grassland | 22 | 7 | 1325 | 28 | 76,600 | 37 | 45,989 | 26 | 125,222 | 17 | 29 | 13 | 3851 | 5 | 253,038 | 176,438 |
| | 4-Shrubland | 7 | 2 | 229 | 5 | 39,079 | 19 | 83,058 | 46 | 54,151 | 7 | 12 | 5 | 2739 | 4 | 179,273 | 96,215 |
| | 5-Agricultural land | 144 | 48 | 1127 | 24 | 21,302 | 10 | 3642 | 2 | 444,860 | 60 | 149 | 66 | 631 | 1 | 471,856 | 26,996 |
| | 6-Barren land | 0 | 0 | 1 | 0 | 168 | 0 | 10 | 0 | 815 | 0 | 0 | 0 | 2 | 0 | 995 | 995 |
| | 7-Forest cover | 12 | 4 | 94 | 2 | 68,455 | 33 | 45,624 | 26 | 104,054 | 14 | 33 | 15 | 68,609 | 90 | 286,882 | 218,273 |
| | **Grand Total** | 302 | | 4677 | | 207,964 | | 178,659 | | 735,418 | | 225 | | 75,908 | | 1,203,153 | |
| | **Expansion** | | | 2781 | | 131,364 | | 95,602 | | 290,558 | | 224 | | 7299 | | | |

**Table A3.** Land use and land cover (LULC) change detection matrix in hectare (ha) and percentage for 2008–2013.

| | LULC Class | LULC 2013 | | | | | | | | | | | | | | Grand Total | Loss |
|---|---|---|---|---|---|---|---|---|---|---|---|---|---|---|---|---|---|
| | | 1 | | 2 | | 3 | | 4 | | 5 | | 6 | | 7 | | | |
| | | ha | % | ha | % | ha | % | ha | % | ha | % | ha | % | ha | % | | |
| LULC 2008 | 1-Built-up area | 2 | 15 | 1 | 0 | 6 | 0 | 7 | 0 | 79 | 0 | 8 | 17 | 2 | 0 | 302 | 102 |
| | 2-Water feature | 2 | 0 | 2414 | 14 | 518 | 1 | 328 | 0 | 1400 | 0 | 0 | 0 | 27 | 0 | 4690 | 2276 |
| | 3-Grassland | 19 | 1 | 9965 | 57 | 39,337 | 39 | 57,231 | 36 | 84,286 | 10 | 3 | 6 | 17,162 | 18 | 208,003 | 168,665 |
| | 4-Shrubland | 2 | 0 | 3069 | 18 | 50,609 | 51 | 78,151 | 49 | 31,863 | 4 | 0 | 0 | 14,989 | 15 | 178,683 | 100,531 |
| | 5-Agricultural land | 1134 | 83 | 1859 | 11 | 8581 | 9 | 19,070 | 12 | 702,576 | 85 | 33 | 73 | 2277 | 2 | 735,530 | 32,954 |
| | 6-Barren land | 2 | 0 | 0 | 0 | 0 | 0 | 1 | 0 | 219 | 0 | 2 | 4 | 0 | 0 | 225 | 223 |
| | 7-Forest cover | 1 | 0 | 77 | 0 | 726 | 1 | 3276 | 2 | 8750 | 1 | 0 | 0 | 63,130 | 65 | 75,960 | 12,830 |
| | **Grand Total** | 1361 | | 17,385 | | 99,778 | | 158,064 | | 829,174 | | 45 | | 97,586 | | 1,203,393 | |
| | **Expansion** | 1161 | | 14,971 | | 60,440 | | 79,913 | | 126,597 | | 43 | | 34,456 | | | |

**Table A4.** Land use and land cover (LULC) change detection matrix in hectare (ha) and percentage for 2013–2018.

| | LULC Class | LULC 2018 | | | | | | | | | | | | | | Grand Total | Loss |
|---|---|---|---|---|---|---|---|---|---|---|---|---|---|---|---|---|---|
| | | 1 | | 2 | | 3 | | 4 | | 5 | | 6 | | 7 | | | |
| | | ha | % | ha | % | ha | % | ha | % | ha | % | ha | % | ha | % | | |
| LULC 2013 | 1-Built-up area | 922 | 20 | 1 | 0 | 0 | 0 | 16 | 0 | 382 | 0 | 40 | 3 | 0 | 0 | 1362 | 440 |
| | 2-Water feature | 8 | 0 | 3,692 | 35 | 2748 | 4 | 8246 | 4 | 2592 | 0 | 0 | 0 | 98 | 0 | 17,384 | 13,692 |
| | 3-Grassland | 17 | 0 | 2202 | 21 | 24,007 | 32 | 63,452 | 30 | 9037 | 1 | 7 | 0 | 1061 | 1 | 99,783 | 75,776 |
| | 4-Shrubland | 171 | 4 | 1316 | 12 | 32,037 | 42 | 75,857 | 36 | 44,801 | 5 | 51 | 4 | 3843 | 5 | 158,076 | 82,219 |
| | 5-Agricultural land | 3536 | 75 | 3,254 | 31 | 11,943 | 16 | 55,117 | 26 | 751,494 | 91 | 1270 | 91 | 2636 | 4 | 829,250 | 77,756 |
| | 6-Barren land | 14 | 0 | 0 | 0 | 0 | 0 | 0 | 0 | 11 | 0 | 20 | 1 | 0 | 0 | 45 | 25 |
| | 7-Forest cover | 28 | 1 | 160 | 2 | 4943 | 7 | 10,863 | 5 | 14,872 | 2 | 6 | 0 | 66,747 | 90 | 97,618 | 30,871 |
| | Grand Total | 4697 | | 10,625 | | 75,677 | | 213,551 | | 823,188 | | 1395 | | 74,385 | | 1,203,518 | |
| | Expansion | 3775 | | 6933 | | 51,670 | | 137,694 | | 71,694 | | 1375 | | 7638 | | | |

**Table A5.** Land use and land cover (LULC) change detection matrix in hectare (ha) and percentage for 1998–2018.

| | LULC Class | LULC 2018 | | | | | | | | | | | | | | Grand Total | Loss |
|---|---|---|---|---|---|---|---|---|---|---|---|---|---|---|---|---|---|
| | | 1 | | 2 | | 3 | | 4 | | 5 | | 6 | | 7 | | | |
| | | ha | % | ha | % | ha | % | ha | % | ha | % | ha | % | ha | % | | |
| LULC 1998 | 1-Built-up area | 37 | 1 | 0 | 0 | 0 | 0 | 0.6 | 0 | 10 | 0 | 1 | 0 | 0 | 0 | 48 | 11 |
| | 2-Water feature | 2 | 0 | 1677 | 16 | 24 | 0 | 193 | 0 | 397 | 0 | 0 | 0 | 9 | 0 | 2302 | 625 |
| | 3-Grassland | 147 | 3 | 2244 | 21 | 31,441 | 42 | 54,617 | 26 | 60,760 | 7 | 64 | 5 | 2425 | 3 | 151,696 | 120,256 |
| | 4-Shrubland | 1150 | 24 | 3365 | 32 | 28,626 | 38 | 912,701 | 43 | 28,709 | 3 | 107 | 8 | 1651 | 2 | 154,878 | 63,607 |
| | 5-Agricultural land | 3005 | 64 | 1914 | 18 | 4747 | 6 | 23,682 | 11 | 499,919 | 61 | 871 | 62 | 1386 | 2 | 535,524 | 35,605 |
| | 6-Barren land | 0 | 0 | 0 | 0 | 0 | 0 | 0 | 0 | 14 | 0 | 0 | 0 | 0 | 0 | 16 | 16 |
| | 7-Forest cover | 357 | 8 | 1365 | 13 | 10,815 | 14 | 43,712 | 20 | 233,232 | 28 | 352 | 25 | 68,868 | 93 | 358,700 | 289,832 |
| | Grand Total | 4697 | | 10,565 | | 75,653 | | 213,476 | | 823,040 | | 1394 | | 74,338 | | 1,203,164 | |
| | Expansion | 4660 | | 8888 | | 44,213 | | 122,205 | | 323,121 | | 1394 | | 5470 | | | |

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
