# Peer review of "Evaluation of Land Use and Land Cover Change and Its Drivers in Battambang Province, Cambodia from 1998 to 2018"

_sustainability, doi:10.3390/su132011170_

Round 1

Reviewer 1 Report

Although standard approaches and methods were used in the study, the results of the work are interesting, reasoned well enough (generally), and practically significant for the studied province of Cambodia. I have several recommendations for improving the content of the manuscript.

  1. The authors reasoned in sufficient detail the identified dynamics of land use/cover by socio-economic drivers during the last decades. Most likely, these drivers were the main ones. However, the influence of natural drivers was not fully described (Section 3.6.4.). The authors did not provide a comparative analysis of the impact of climate change on the expansion of cropland in the lowland and mountainous parts of the province (floods, changes in the groundwater level due to increased evaporation (evapotranspiration?) with increasing temperatures, changes in precipitation in the lowlands and the mountains, etc.). Other environmental factors are not as dynamic, but they could also determine the direction of this expansion. For example, the soil. How has soil fertility changed in recent decades in the lowland part of the province? How attractive are the soil cover resources in the mountains of the province for the expansion of cropland? Without this analysis, the findings of your research seem one-sided. You are planning to publish this manuscript in the journal Sustainability. But the sustainable development of any territory is not determined only by human activity!
  2. You have analyzed the differences in land-cover changes in the lowland and mountainous parts of Battambang province. In this regard, in subsection 2.1., it is necessary to provide a more detailed description of environmental differences between these two parts (differences in the area, the main characteristics of the relief (topography), surface rocks, soils, temperature, precipitation, population, etc.). It is desirable to do this in the form of a comparative table. Otherwise, the analysis carried out in the lowlands and mountains of the province is difficult to perceive. Moreover, if the differences are not significant, then why did you analyze the differences in land-cover dynamics in these two parts? Convince the reader that these environmental differences are significant.
  3. The symbols in the formulas in lines 194 and 198 and the symbols in the explanations for these formulas must be strictly identical. Check it out carefully. There are inconsistencies.
  4. What are the units of measure for Tables 3? %?
  5. Lines 247-272. The content of these lines is a detailed description of Table 5. Is this duplication necessary? This routine description looks boring. Please shorten this part of the text. Leave only the most important. Something similar is in subsection 3.5.

Author Response

Dear Reviewer 1

We are thankful for the edits of reviewer 1. We have fulfilled and revised the manuscript based on the valuable comments from the reviewer 1 accordingly. Some paragraphs and sentences are added in the revised manuscript.  We have included a REVISED VERSION in word that shows the revision we used TRACK CHANGES.  Please kindly see the attachment.

Thank you very much for your kind acceptance of our response in advance.

Best Regards,

Authors

Reviewer 2 Report

Dear authors,

The manuscript is very good, the authors evaluate the Land Use and Land Cover Change and its Drivers in Battambang Province, Cambodia from 1998 to 2018. It is an interesting and great contribution to the scientific community, however, the introduction and conclusions of the paper should be improved. You can review the suggestions in the attached document.

Author Response

Dear Reviewer 2;

 We have fulfilled and revised the manuscript based on the valuable comments from the reviewer 2 accordingly. Some paragraphs and sentences are added in the revised manuscript.   We have included a REVISED VERSION in word that shows the revision we used TRACK CHANGES.  Please kindly see our response in the attachment.

We hope that you are satisfied with our revisions and responses and we also thank you very much for your expending valuable time to comment on our manuscript.

Best regards,

Authors

Reviewer 3 Report

It is very interesting to study land use changes in particular areas. Cambodia has been threatened by war and is one of the world's less developed countries. Agriculture is one of the sources of sustainable livelihoods for farmers. Therefore, the study of agricultural land use is very important. The authors have paid good attention to these issues and the paper is well organized. In summary, I recommend publishing it directly.

Author Response

Dear Reviewer 3;

We would like to thank you very much for your kind reviewing our manuscript and allow us publish directly.

Best regards,

Authors

Reviewer 4 Report

Very interesting study on LULC change and drivers in Cambodia. Introduction could lay down clear objectives of this study.

What is the source of the referenced or ground truth data?

Provide explanation of a low Kappa in yr. 2003.

Author Response

Dear Reviewer 4;

 We have fulfilled and revised the manuscript based on the valuable comments from the reviewer 4 accordingly. Source of reference data and ground truth are added in the revised manuscript. We have explained the low Kappa in yr 2003 in only Response letter.

We hope that you are satisfied with our revisions and responses and we also thank you very much for your expending valuable time to comment on our manuscript.

Best regards,

Authors

Round 2

Reviewer 1 Report

Dear Authors,

  1. Put in order the numbering of all figures in the manuscript.
  2. Remove the outer frame in:
    Figure 5. Detailed dynamics of LULC change in the upland ...
    Figure 4 (?). Type of Road in Battambang province ...
  3. Table 1. The minimum elevation in the province corresponds to the average water level of Tonle Sap Lake (0.5 m a.s.l.). Within the upland part of the province, it cannot be 0 m in any way. Would you please recalculate it? Moreover, what does "Topographic" mean? It would be correct to write "Elevation". Check the rest of the text.

Author Response

Dear Reviewer 1 (Round 2)

Thank you for giving us the opportunity to revise our manuscript. We appreciate comments suggested by Reviewer 1 (round 2). These suggestions improved the quality of our manuscript. We are submitting revised version of our manuscript with track changes. We have highlighted the revised text in red colored font.

Thank you very much for your kind acceptance of our response in advance.

Best Regards,

Authors

This manuscript is a resubmission of an earlier submission. The following is a list of the peer review reports and author responses from that submission.